# Skip the Steps: Data-Free Consistency Distillation for Diffusion-based Samplers

## Abstract

Sampling from probability distributions is a fundamental task in machine learning and statistics. However, most existing algorithms require numerous iterative steps to transform a prior distribution into high-quality samples, resulting in high computational costs and limiting their practicality in time-constrained and resource-limited environments. In this work, we propose *consistency samplers*, a novel class of samplers capable of generating high-quality samples in a single step. Our method introduces a new consistency distillation algorithm for diffusion-based samplers, which eliminates the need for data or full trajectory integration. By utilizing incomplete sampling trajectories and noisy intermediate representations along the diffusion process, we efficiently learn a direct one-step mapping from any state to its corresponding terminal state in the target distribution. Moreover, our approach enables few-step sampling, allowing users to flexibly balance compute costs and sample quality. We demonstrate the effectiveness of consistency samplers across multiple benchmark tasks, achieving high-quality results with one-step or few-step sampling while significantly reducing the sampling time compared to existing samplers. For instance, our method is 100-200x faster than prior diffusion-based samplers while having comparable sample quality.

## 1 Introduction

Sampling from an unnormalized target distribution $\rho \propto p_{\text{target}}$ without access to data samples is a fundamental challenge across various domains, including machine learning (Neal, 1995; Hernández-Lobato & Adams, 2015), statistics (Neal, 2001; Andrieu et al., 2003), physics (Wu et al., 2019; Albergo et al., 2019), chemistry (Frenkel & Smit, 2002; Hollingsworth & Dror, 2018), and many other fields involving probabilistic models.

Many existing sampling algorithms are inherently iterative, with the accuracy of the final samples depending heavily on the number of steps. For example, Markov chain Monte Carlo (MCMC) methods rely on iteratively generating samples through a Markov chain that converges to the target distribution (MacKay, 2003; Robert, 1995). Similarly, diffusion-based samplers frame sampling as a stochastic optimal control problem, transforming samples from a simple prior distribution into the target distribution by iteratively solving a controlled stochastic differential equation (SDE) (Zhang & Chen, 2022; Vargas et al., 2023; Berner et al., 2024; Zhang et al., 2024; Richter & Berner, 2024). However, these iterative samplers often suffer from slow mixing and require hundreds or even more steps to converge, making them impractical for use in large models and resource-limited scenarios.

In this work, we propose a novel class of samplers, *consistency samplers* (CS), that can generate high-quality samples in just a single step. A comparison between CS and existing iterative samplers is shown in Figure 1. To achieve one-step sampling, our method introduces a new distillation algorithm for diffusion-based samplers, inspired by the idea of consistency models (CM) (Song et al., 2023). Unlike CMs, our approach does not require access to data or the generation of full sampling trajectories. Instead, it leverages intermediate noisy representations to learn the consistency function, significantly reducing the computational overhead of the training process. In our numerical experiments, we demonstrate the effectiveness of consistency samplers across multiple benchmark tasks, achieving high-quality results with one-step or few-step sampling, and drastically reducing the sampling time compared to existing methods. In summary, our contributions are as follows:

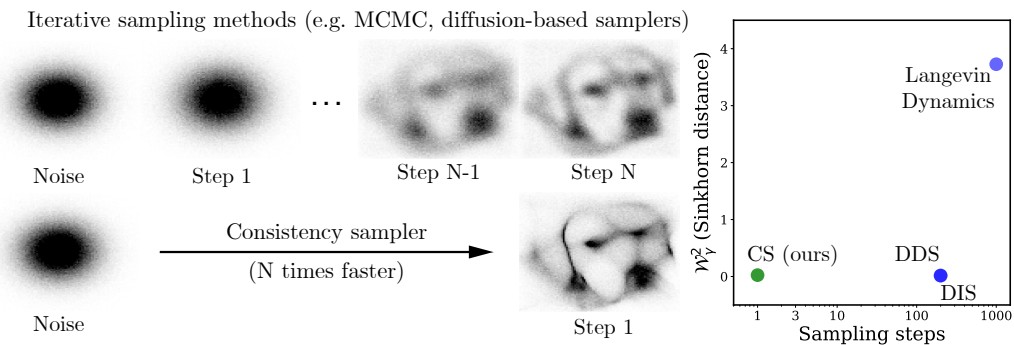

Figure 1: The proposed consistency sampler (CS) achieves high-quality sampling in just one step, significantly accelerating the sampling process compared to methods like MCMC (e.g. Langevin Dynamics) and diffusion-based sampling (e.g. DDS, DIS), which require numerous steps to gradually generate samples.

- We introduce consistency samplers, a new class of samplers that can generate high-quality samples in one or a few steps from complex unnormalized distributions. Our distillation training algorithm is computationally efficient, requiring only incomplete sampling trajectories from diffusion-based samplers and eliminating the need for pre-collected data.
- We provide a theoretical analysis of the proposed consistency distillation training objective, establishing guarantees on the convergence and correctness of the consistency sampler under our training framework.
- We empirically demonstrate that our consistency samplers perform effectively on standard sampling benchmarks, achieving high-quality results in both one-step and few-step sampling tasks. Our approach accelerates sampling by 100-200x and reduces the neural network size by half compared to previous diffusion-based samplers, all while maintaining comparable sample quality.

## 2 RELATED WORK

**Iterative sampling methods.** Markov chain Monte Carlo (MCMC) is commonly used for sampling unnormalized distributions. The core idea is to construct a Markov chain whose equilibrium distribution matches the desired target distribution (Brooks et al., 2012). Popular MCMC algorithms include Metropolis-Hasting (Metropolis et al., 1953; Hastings, 1970), Gibbs sampling (Geman & Geman, 1984), and Langevin dynamics (Rossky et al., 1978; Parisi, 1981). Instead of propagating a single sample, sequential Monte Carlo (SMC) methods propagate a population of particles through a sequence of intermediate distributions (Doucet et al., 2001). An example is annealed importance sampling, which transforms a simple distribution into the target distribution using annealed intermediate distributions and importance weights (Neal, 2001).

The classical Schrödinger bridge problem (Schrödinger, 1931; 1932) seeks the most likely stochastic process that transports one distribution to another consistently with a pre-specified Brownian motion. The sampling problem can then be framed as an optimal control problem, where a controlled SDE is used to evolve samples from an initial distribution through the Schrödinger bridge to the target distribution (Tzen & Raginsky, 2019; Vargas et al., 2022; Zhang & Chen, 2022). This approach motivates the study of diffusion processes as samplers Geffner & Domke (2023); Vargas et al. (2023); Richter & Berner (2024); Zhang et al. (2024); Phillips et al. (2024).

Key to MCMC, SMC, and diffusion-based samplers is their iterative nature, where each method progressively refines samples through a series of transformations or updates to more accurately represent the target distribution. Our work asks whether it is possible to skip the iterative refinement process by learning to directly map the initial distribution to the target.

**Accelerating strategies for sampling.** Robert et al. (2018) surveys various techniques to improve MCMC efficiency, including Hamiltonian Monte Carlo, which leverages the geometry of the target

distribution for more effective sampling (Duane et al., 1987; MacKay, 2003; Brooks et al., 2012; Chen et al., 2014). To reduce costs on large datasets, subsampling MCMC methods (Bardenet et al., 2017; Andrieu & Roberts, 2009; Zhang & De Sa, 2019; Zhang et al., 2020b) and stochastic gradient MCMC methods (Welling & Teh, 2011; Chen et al., 2014; Zhang et al., 2020a;c) have been developed. These approaches are orthogonal to our method since they reduce the cost per step but remain fundamentally iterative in nature.

Amortized inference, on the other hand, shifts the computational cost to a training phase, resulting in a sampler that is faster at test time (Gershman & Goodman, 2014). For instance, amortized MCMC (Li et al., 2017) distills an MCMC sampler by training a student model to mimic the sample after $T$-step MCMC transitions. Most amortized inference methods rely on simulation-based training, where a teacher sampler generates data during training. GFlowNets (Bengio et al., 2021; 2023) focus on sampling complex composite objects by sequentially composing their elements. While GFlowNets amortize the computational challenges of lengthy stochastic searches and mode-mixing during training, their sampling process remains sequential, as objects are constructed step-by-step through a series of constructive steps. In contrast to amortized MCMC, our method only requires generating incomplete samples during training and enables single-step sampling, unlike the sequential sampling process of GFlowNets' generative policy.

**Diffusion generative models.** In contrast to diffusion-based samplers, diffusion generative models rely on direct access to data from the target distribution and progressively perturb this data toward noise via a diffusion process (Sohl-Dickstein et al., 2015; Song & Ermon, 2019; Ho et al., 2020; Song et al., 2021b). The generative process learns to reverse this diffusion through denoising score matching (Hyvärinen, 2005; Vincent, 2011). Several strategies have been proposed to accelerate the generation process of diffusion generative models. For example, faster solvers (Song et al., 2021a; Nichol & Dhariwal, 2021; Jolicoeur-Martineau et al., 2021; Karras et al., 2022) reduce the number of reverse iterations from hundreds or thousands to just tens. Additionally, knowledge distillation techniques can further minimize the number of steps, allowing for single-step or few-step generation (Salimans & Ho, 2022; Song et al., 2023). In this work, we extend ideas from distillation techniques for diffusion generative models to diffusion-based samplers to design an efficient, single-step sampler.

## 3 PRELIMINARIES: DIFFUSION-BASED SAMPLING

Diffusion-based samplers are controlled stochastic processes that gradually transform samples from a simple prior distribution $\mathbf{x}_0 \sim p_{\text{prior}}$ into approximate samples from the target distribution $\mathbf{x}_T \sim p_{\text{target}}$ by evolving forward in time $t \in [0, T]$:

$$\mathrm{d}\mathbf{x}_t^u = (\mu(\mathbf{x}_t^u, t) + \sigma(t)u(\mathbf{x}_t^u, t))\,\mathrm{d}t + \sigma(t)\,\mathrm{d}\mathbf{w}_t, \tag{1}$$

where $\mathbf{w}$ is a standard Brownian motion, $\mu$ is the drift term, $\sigma$ is the diffusion coefficient, and $u$ is a control term that adjusts the drift.

The objective is to find $u$ such that Eq. (1) approximates the reverse-time process of an inference diffusion that adds noise to samples drawn from the target distribution:

$$\mathrm{d}\mathbf{y}_t^v = (\mu(\mathbf{y}_t^v, t) + \sigma(t)v(\mathbf{y}_t^v, t))\,\mathrm{d}t + \sigma(t)\,\mathrm{d}\mathbf{w}_t. \tag{2}$$

where $v(\mathbf{y}_t^v, t) = \sigma^\top(t)\nabla \log p_{\mathbf{y}_t^v}(\mathbf{y}_t)$ (Anderson, 1982).

By ensuring that $\mathbf{y}_0^v \sim p_{\text{prior}}$ and $u = v$, one can achieve $p_{\mathbf{x}^u} = p_{\mathbf{y}^v}$, and thus $\mathbf{x}_T^u \sim p_{\text{target}}$. However, directly computing the score $\nabla \log p_{\mathbf{y}^v}$ is intractable, and we assume that no dataset from $p_{\text{target}}$ is available to approximate it.

Let $\mathbb{P}_{\mathbf{x}^u}$ denote the path space measure corresponding to the process defined by Eq. (1), and let $\mathbb{P}_{\mathbf{y}^v}$ denote the path space measure of the process defined by Eq. (2). Further, let $\mathcal{U} \subset C(\mathbb{R}^d \times [0, T], \mathbb{R}^d)$ represent the space of admissible controls. Diffusion-based samplers seek to find an optimal control $u^*$ that minimizes the divergence between the path measures of the generative and time-reversed inference processes:

$$u^* \in \arg\min_{\mathcal{U}} D(\mathbb{P}_{\mathbf{x}^u} \| \mathbb{P}_{\mathbf{y}^v}), \tag{3}$$

where $D(\cdot\|\cdot)$ is an appropriate divergence measure (e.g., Kullback-Leibler (KL) divergence) between the two path distributions.

In practice, one then generates samples by simulating $\mathbf{x}^{u^*}$ using the Euler-Maruyama integrator:

$$\mathbf{x}_{t+\Delta t} = \mathbf{x}_t + (\mu(\mathbf{x}_t, t) + \sigma(t)u^*(\mathbf{x}_t, t))\,\Delta t + \sigma(t)\Delta\mathbf{w}_t, \quad \Delta\mathbf{w}_t \sim \mathcal{N}(0, \Delta t\boldsymbol{I}) \tag{4}$$

where $\Delta t$ is the step size. The smaller $\Delta t$ is, the more accurate the approximation becomes, but this also increases the number of required steps $N$, and thus, the computational cost.

## 4 CONSISTENCY SAMPLER

In this section, we introduce consistency samplers, a method for distilling diffusion-based samplers into single-step samplers.

### 4.1 PARAMETERIZATION

We propose distilling a diffusion process induced by a control function $u$, which satisfies the problem in Eq. (3), into what we call a consistency sampler. Given $u$, the consistency sampler learns a deterministic consistency function $f : (\mathbf{x}_t^u, t) \mapsto \mathbf{x}_T^u$, which maps any intermediate state of a path directly to its terminal state. As a result, one-step sampling becomes feasible from any point in time, in particular from the initial state.

To ensure that the learned consistency function outputs the correct terminal state, we parameterize the consistency sampler such that the consistency function is the identity $f(\mathbf{x}_T^u, T) = \mathbf{x}_T^u$ at the terminal time. Following Song et al. (2023), the consistency sampler is parameterized as follows:

$$f_{\boldsymbol{\theta}}(\mathbf{x}_t^u, t) = c_{\text{skip}}(t)\mathbf{x}_t^u + c_{\text{out}}(t)F_{\boldsymbol{\theta}}(\mathbf{x}_t^u, t), \tag{5}$$

where the coefficients $c_{\text{skip}}(t)$ and $c_{\text{out}}(t)$ are such that $c_{\text{skip}}(T) = 1$ and $c_{\text{out}}(T) = 0$, ensuring that the output is equal to the terminal state. Here, $F_{\boldsymbol{\theta}}$ is a free-form neural network, and its architecture can be borrowed from prior diffusion-based samplers.

To train the consistency sampler, we aim to ensure that the learned function provides consistent mappings between adjacent points along the diffusion trajectory. Specifically, we minimize the difference between the outputs of the consistency function applied to the states of two consecutive time steps, $\mathbf{x}_{t_n}^u$ and $\mathbf{x}_{t_{n+1}}^u$, in a given time discretization.

Consistency distillation of diffusion generative models rely on a direct access to samples from $p_{\text{target}}$ to learn the consistency function (Song et al., 2023). In contrast, our approach assumes that we do not have access to data, and generating a dataset of samples from the target distribution using a pre-trained sampler is computationally expensive.

### 4.2 EFFICIENT INTERMEDIATE CONSECUTIVE STATES GENERATION

In practice, the SDEs commonly used in diffusion-based samplers often have linear drift terms of the form $\mu(\mathbf{x}_t, t) = \mu(t)\mathbf{x}_t$. This is true for widely used SDEs such as the variance exploding and variance preserving SDEs (Song et al., 2021b). In such cases, the backward perturbation kernels, which describe the transition from $\mathbf{x}_T$ to $\mathbf{x}_t$, are known to follow Gaussian transitions:

$$P_B(\mathbf{x}_t|\mathbf{x}_T) = \mathcal{N}(\mathbf{x}_t; s(t)\mathbf{x}_T,\ s(t)^2 g(t)^2\mathbf{I}), \tag{6}$$

where

$$s(t) = \exp\left(\int_0^{T-t} \mu(\xi)\,\mathrm{d}\xi\right), \quad \text{and} \quad g(t) = \sqrt{\int_0^{T-t} \frac{\sigma(\xi)^2}{s(\xi)^2}\,\mathrm{d}\xi}.$$

See Eq. (29) in Song et al. (2021b), and Appendix B of Karras et al. (2022).

A straightforward approach to learn the consistency function would be to generate approximate samples $\hat{\mathbf{x}}_T^u$ by fully integrating the diffusion process from noise, and then applying the backward perturbation kernel to obtain consecutive intermediate states $\hat{\mathbf{x}}_{t_n}^u$ and $\hat{\mathbf{x}}_{t_{n+1}}^u$. While this method allows for the direct application of the consistency techniques from Song et al. (2023); Song & Dhariwal (2023), it is inefficient as it requires fully integrating the process for every training iteration.

We propose a more efficient method that avoids the need for full integration. Starting with an initial sample $\mathbf{x}_0 \sim p_{\text{prior}}$, we randomly sample a timestep $t_n$ from a predefined time discretization and

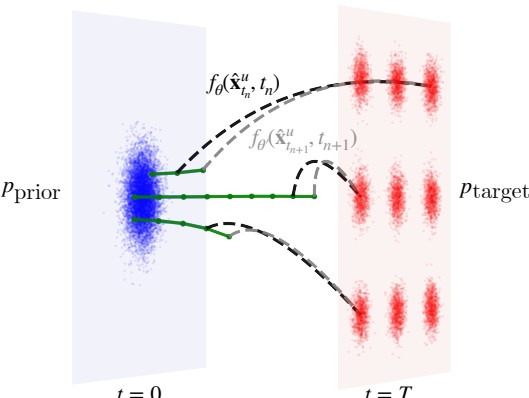

Figure 2: Consistency samplers are trained to map consecutive points (indicated by the black and gray dashed curves) along the partially integrated trajectory of the PF ODE (represented by the green curves), bypassing the need for fully integrated trajectories.

simulate the forward process only up to $t_{n+1}$. This provides the intermediate states $\hat{\mathbf{x}}^u_{t_n}$ and $\hat{\mathbf{x}}^u_{t_{n+1}}$ directly, without needing to run the entire process up to $T$, thus reducing the training time. The proposed training procedure is illustrated in Figure 2.

### 4.3 PROBABILITY FLOW ODE FOR DETERMINISTIC TRANSITIONS

When simulating the forward SDE (Eq. (1)), the transition between two states is stochastic due to the randomness introduced by the Brownian motion. This stochasticity creates challenges for learning the consistency function, as the probabilistic nature of state transitions induces ambiguity in the mapping. Specifically, the same intermediate state $\mathbf{x}^u_t$ can correspond to multiple potential future paths, complicating the task of learning a unique and consistent mapping from $\mathbf{x}^u_{t_n}$ and $\mathbf{x}^u_{t_{n+1}}$ to $\mathbf{x}^u_T$.

Fortunately, for all diffusion processes, there exists a corresponding deterministic process whose trajectories share the same marginal probability densities as the original SDE (Song et al., 2021b). This deterministic process is governed by an ordinary differential equation (ODE), commonly referred to as the probability flow ODE (PF ODE). The PF ODE corresponding to the forward generative SDE (Eq. (1)) is:

$$\mathrm{d}\mathbf{x}^u_t = \left( \mu(\mathbf{x}^u_t, t) + \frac{1}{2}\sigma(t)u(\mathbf{x}^u_t, t) \right) \mathrm{d}t. \tag{7}$$

By leveraging the PF ODE, we can obtain deterministic consecutive points $\hat{\mathbf{x}}^u_{t_n}$ and $\hat{\mathbf{x}}^u_{t_{n+1}}$ for training the consistency function, thus avoiding the stochasticity challenges posed by the SDE. When simulating the pre-trained diffusion-based sampler during training, we therefore use the PF ODE (Eq. (7)) instead of the SDE (Eq. (1)).

### 4.4 TRAINING OBJECTIVE AND THEORETICAL GUARANTEES

Given a time discretization $0 < t_1 < \cdots < t_N = T$, the consistency sampler is trained to minimize the difference between the outputs of the consistency function at $\hat{\mathbf{x}}^u_{t_n}$ and $\hat{\mathbf{x}}^u_{t_{n+1}}$, obtained by integrating the PF ODE (Eq. (7)) from $t_0$ to $t_{n+1}$.

The training loss is formulated as:

$$\mathcal{L}(\boldsymbol{\theta}, \boldsymbol{\theta}'; u) := \mathbb{E}\left[ \lambda(t_n)d(f_{\boldsymbol{\theta}'}(\hat{\mathbf{x}}^u_{t_{n+1}}, t_{n+1}), f_{\boldsymbol{\theta}}(\hat{\mathbf{x}}^u_{t_n}, t_n)) \right] \tag{8}$$

where $\boldsymbol{\theta}' \leftarrow \mathrm{stopgrad}(\boldsymbol{\theta})$, and $\lambda(\cdot)$ is a positive weighting function that controls the contribution of each time step to the loss, and $d(\cdot, \cdot)$ is a distance metric. The training procedure is outlined in Algorithm 1.

---

**Algorithm 1** Data-free consistency sampler training

---

**Input** model parameters $\boldsymbol{\theta}$, control $u$, learning rate $\eta$, distance metric $d(\cdot, \cdot)$, loss weighting $\lambda(\cdot)$
$\boldsymbol{\theta}' \leftarrow \boldsymbol{\theta}$
**repeat**
    Sample $\mathbf{x}_0 \sim p_{\text{prior}}$ and $n \sim \mathcal{U}\{1, N - 1\}$
    Sample $\hat{\mathbf{x}}_{t_{n+1}}^u$ and $\hat{\mathbf{x}}_{t_n}^u$ by simulating Eq. (7) from $\mathbf{x}_0$ to $\hat{\mathbf{x}}_{t_{n+1}}^u$ using $u$
    $\mathcal{L}(\boldsymbol{\theta}, \boldsymbol{\theta}'; u) \leftarrow \lambda(t_n) d(f_{\boldsymbol{\theta}'}(\hat{\mathbf{x}}_{t_{n+1}}^u, t_{n+1}), f_{\boldsymbol{\theta}}(\hat{\mathbf{x}}_{t_n}^u, t_n))$
    $\boldsymbol{\theta} \leftarrow \boldsymbol{\theta} - \eta \nabla_{\boldsymbol{\theta}} \mathcal{L}(\boldsymbol{\theta}, \boldsymbol{\theta}'; u)$
    $\boldsymbol{\theta}' \leftarrow \text{stopgrad}(\boldsymbol{\theta})$
**until** convergence

---

---

**Algorithm 2** Multi-step sampling from a consistency sampler

---

**Input** Consistency sampler $f_{\boldsymbol{\theta}}(\cdot, \cdot)$, sequence of timesteps $t_1 < \cdots < t_n$
Sample $\mathbf{x}_0 \sim p_{\text{prior}}$
$\mathbf{x}_T \leftarrow f_{\boldsymbol{\theta}}(\mathbf{x}_0, 0)$
**for** $i = 1$ **to** $n$ **do**
    Sample $\mathbf{x}_{t_i}$ from Eq. (6)
    $\mathbf{x}_T \leftarrow f_{\boldsymbol{\theta}}(\mathbf{x}_{t_i}, t_i)$
**end for**
**Return** $\mathbf{x}_T$ as the generated sample.

---

Next, we provide an asymptotic analysis of the error between the learned consistency sampler and the true consistency function induced by the pre-trained control and the PF ODE (Eq. (7)) when optimizing the loss in Eq. (8).

**Theorem 1.** *Let $\boldsymbol{f}_{\boldsymbol{\theta}}(\mathbf{x}_t, t)$ be a consistency sampler parameterized by $\boldsymbol{\theta}$, and let $\boldsymbol{f}(\mathbf{x}_t, t; u)$ denote the consistency function of the PF ODE defined by the control $u$. Assume that $\boldsymbol{f}_{\boldsymbol{\theta}}$ satisfies a Lipschitz condition with constant $L > 0$, such that for all $t \in [0, T]$ and for all $\mathbf{x}_t, \mathbf{y}_t$,*

$$\|\boldsymbol{f}_{\boldsymbol{\theta}}(\mathbf{x}_t, t) - \boldsymbol{f}_{\boldsymbol{\theta}}(\mathbf{y}_t, t)\|_2 \le L \|\mathbf{x}_t - \mathbf{y}_t\|_2.$$

*Additionally, assume that for each step $n \in \{1, 2, \ldots, N - 1\}$, the ODE solver called at $t_n$ has a local error bounded by $O((t_{n+1} - t_n)^{p+1})$ for some $p \ge 1$.*

*If, additionally, $\mathcal{L}(\boldsymbol{\theta}, \boldsymbol{\theta}; u) = 0$, then:*

$$\sup_{n, \mathbf{x}_{t_n}} \|\boldsymbol{f}_{\boldsymbol{\theta}}(\mathbf{x}_{t_n}, t_n) - \boldsymbol{f}(\mathbf{x}_{t_n}, t_n; u)\|_2 = O((\Delta t)^p),$$

*where $\Delta t := \max_{n \in \{1, 2, \ldots, N-1\}} |t_{n+1} - t_n|$.*

*Proof.* We provide a proof in Appendix B. $\qquad\square$

If the consistency sampler achieves zero loss, Theorem 1 implies that, under regularity conditions, the estimated consistency sampler can become arbitrarily accurate as the step size of the ODE solver decreases, ensuring the learned model closely approximates the true consistency function.

## 4.5 SAMPLING FROM CONSISTENCY SAMPLERS

With a well-trained consistency sampler $f_{\boldsymbol{\theta}}(\cdot, \cdot)$, we can generate approximate samples from the target distribution in a single step by first sampling from the prior distribution $\mathbf{x}_0 \sim p_{\text{prior}}$, and then evaluating the consistency function $f_{\boldsymbol{\theta}}(\mathbf{x}_0, 0)$.

We can also further refine this generated sample by performing multiple denoising and noise addition steps using the backward perturbation kernel from Eq. (6), akin to the consistency models distilled from diffusion generative models. The multi-step sampling procedure is outlined in Algorithm 2.

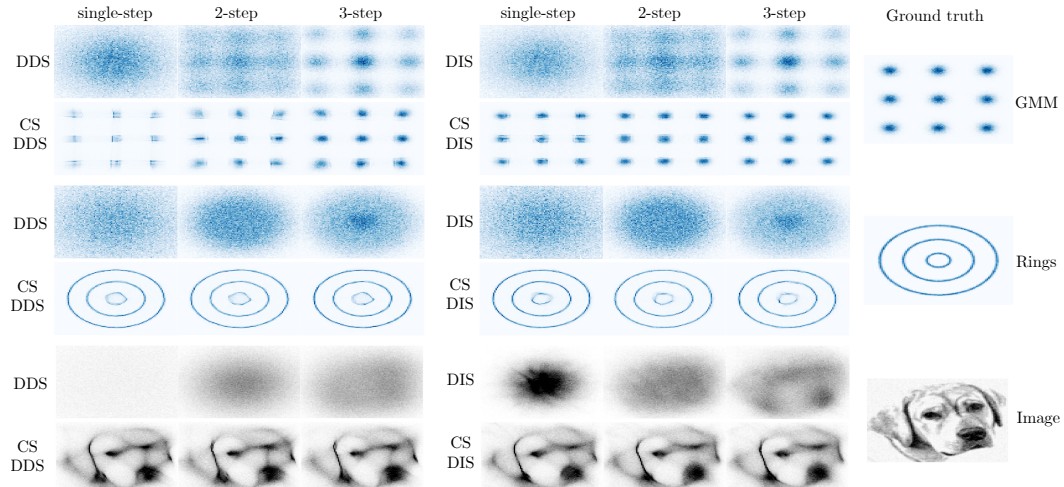

Figure 3: Comparison of samples generated with one, two, and three steps by the consistency sampler (CS), time-reversed diffusion sampler (DIS), and denoising diffusion sampler (DDS) across GMM, rings, and image targets. CS consistently produces sharper results and successfully recovers all modes of the target distributions.

## 5 NUMERICAL EXPERIMENTS

In this section, we empirically evaluate the performance of the proposed consistency sampler, trained using Algorithm 1. The control $u_{\boldsymbol{\theta}}$ is modeled as a neural network, which is pre-trained using either the denoising diffusion sampler (DDS) (Vargas et al., 2023) or the time-reversed diffusion sampler (DIS) (Berner et al., 2024). Both DDS and DIS implementations rely on the PIS-GRAD architecture introduced by Zhang & Chen (2022), where the control is:

$$u_{\boldsymbol{\theta}}(\mathbf{x}_t, t) = \mathrm{NN}_{1;\boldsymbol{\theta}}(\mathbf{x}_t, t) + \mathrm{NN}_{2;\boldsymbol{\theta}}(t) \times \nabla \log p_{\mathrm{target}}(\mathbf{x}_t),$$

with $\mathrm{NN}_{1;\boldsymbol{\theta}}$ and $\mathrm{NN}_{2;\boldsymbol{\theta}}$ representing two neural networks. Across all experiments, we use a two-layer architecture with 64 hidden units each for both networks. The training objectives of DDS and DIS are presented in Appendix A.

The training cost of CS is less then that of the denoising diffusion sampler (DDS) and the time-reversed diffusion sampler (DIS). In DDS and DIS, the controlled process appears directly in the training objective (see equations 10 and 11). Unlike diffusion models that use the denoising score matching objective and can resort to Monte Carlo approximations (Hyvärinen, 2005; Song & Ermon, 2019), DDS and DIS require full trajectory simulation during training. Similarly, CS requires trajectory simulation during training; however, CS integrates only partial trajectories up to a random timestep. This approach saves approximately 50% of the training time for a fixed number of training iterations.

In our parameterization of the consistency sampler (Section 4.1), we initialize the network $F_{\boldsymbol{\theta}}$ in Eq. (5) with $\mathrm{NN}_{1;\boldsymbol{\theta}}(\mathbf{x}_t, t)$. As a result, the consistency sampler requires roughly half the number of parameters compared to DIS and DDS, thereby reducing both the computational cost of a forward pass through the model and the memory requirements.

In all of our experiments, DDS and DIS follow a variance-preserving SDE (Song et al., 2021b) with a Gaussian prior, and are trained using the log-variance divergence (Richter & Berner, 2024) with 200 diffusion steps to solve the optimal control problem of Eq. (3). The consistency sampler is trained with 18 diffusion steps, using $\lambda(t) = 1$ and the L2-norm in the loss Eq. (8).

## 5.1 BENCHMARK TARGETS

**Gaussian mixture model (GMM):** The target distribution for the GMM is defined as:

$$\rho(\mathbf{x}) = \frac{1}{m} \sum_{i=1}^{m} \mathcal{N}(\mathbf{x}; \boldsymbol{\mu}_i, \boldsymbol{\Sigma}_i)$$

where $m = 9$, the covariance matrix $\sigma_i = 0.3\boldsymbol{I}$, and the means $(\mu_i)_{i=1}^{9}$ are positioned at the points in $\{-5, 0, 5\}^2$. This creates a mixture of nine 2D Gaussian components.

**Double well (DW):** A common challenge in molecular dynamics is sampling from the stationary distribution of a Langevin dynamic system. In our case, we consider a $d$-dimensional double well potential, characterized by the following (unnormalized) density:

$$\rho(\mathbf{x}) = \exp\left(-\sum_{i=1}^{m}(x_i^2 - \delta) - \frac{1}{2}\sum_{i=m+1}^{d} x_i^2\right).$$

Here, $m \in \mathbb{N}$ represents the number of double wells, and $\delta \in (0, \infty)$ is a separation parameter controlling the distance between the wells. The first $m$ dimensions contribute to the double well potential, while the remaining dimensions follow a simple Gaussian form.

**Rings:** The rings distribution is a two-dimensional mixture of concentric rings centered at the origin, with each ring having a different radius. Each ring is modeled as a distribution concentrated around a specific radius with some Gaussian perturbation. The density is

$$\rho(\mathbf{x}) = \exp\left(-\min_i \frac{1}{2\sigma^2}\left(\|\mathbf{x}\| - r_i\right)^2\right)$$

where $r_i$ is the radius of the $i$-th ring, $\sigma$ is a parameter controlling the scale of the Gaussian perturbation around each ring.

**Image:** We use a normalized grayscale image to create a two-dimensional probability density, following the setup from Wu et al. (2020).

## 5.2 DISCUSSION

Figure 3 presents a qualitative comparison of samples generated by CS, DIS, and DDS using one, two, and three steps, across the GMM, rings, and image benchmarks.

One critical observation from Figure 3 is the clear limitation of DIS and DDS in generating high-quality samples with a limited number of network function evaluations (NFEs), likely due to the large step sizes in Euler-Maruyama integration, which introduce significant approximation errors. As a result, DIS and DDS samples display poor mode coverage and lack the sharpness compared to the ground truth distribution. Even with a single step, CS is able to capture the modes of the target distribution, delivering sharper distributions and more accurate samples.

Figure 4 displays the Sinkhorn distance (Cuturi, 2013) between generated samples and the ground truth distribution as a function of NFEs, ranging from 1 to 10. This plot corroborates the findings from Figure 3, clearly demonstrating the superior performance of CS in both single-step and few-step generation tasks. DDS and DIS exhibit significantly higher Sinkhorn distances, indicating that these methods struggle to accurately approximate the target distribution when sampling with few steps.

As the NFEs increase, the performance gap between DDS, DIS, and consistency samplers narrows, suggesting that the advantage of CS diminishes when enough steps are taken. However, this improvement in DDS and DIS comes at the cost of increased computational resources, as they require more NFEs to match the performance that CS achieves with fewer steps.

Table 1 presents the Sinkhorn distances between samples from the ground truth distributions and the samples generated by DDS, DIS, and their CS counterparts. At NFE=200, both DDS and DIS achieve low Sinkhorn distances across all datasets, which aligns with prior findings that given enough function evaluations, both methods can closely match the ground truth distribution.

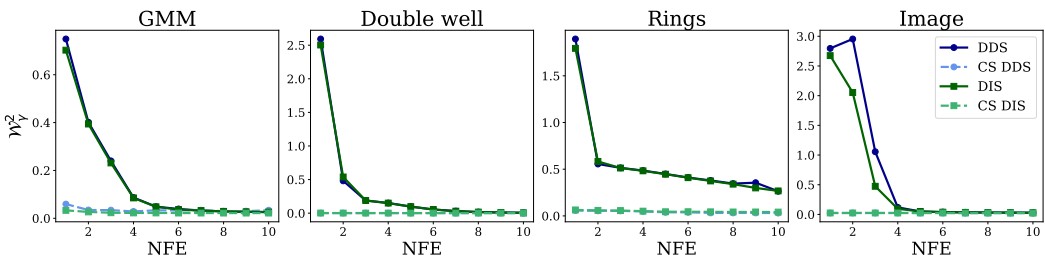

Figure 4: Sinkhorn distance $\mathcal{W}_\gamma^2$ as a function of the number of network function evaluations (NFE) for 2 dimensional tasks. The consistency samplers (CS) achieve lower Sinkhorn distances compared to DDS and DIS in the few-step sampling regime

Table 1: Sinkhorn distances $\mathcal{W}_\gamma^2$ between samples from the ground truth distribution and samples from DDS, DIS, and their respective distilled consistency samplers (CS), with varying number of network function evaluations (NFE). CS achieves results comparable to 200-step DIS and DDS while being 100 to 200 times faster.

| Method | NFE | GMM | Rings | Image | DW shift (d=2) | DW (d=100) |
|--------|-----|-----|-------|-------|----------------|------------|
| DDS | 200 | 0.0205 | 0.0180 | 0.0162 | 0.0012 | 10.9340 |
| DIS | 200 | 0.0206 | 0.0178 | 0.0163 | 0.0012 | 10.9658 |
| DDS | 2 | 0.4012 | 0.5556 | 2.9546 | 0.4811 | 30.5513 |
| DIS | 2 | 0.3939 | 0.5827 | 2.0533 | 0.5388 | 20.7306 |
| CS DDS | 2 | 0.0347 | **0.0545** | **0.0240** | **0.0013** | **9.4402** |
| CS DIS | 2 | **0.0266** | 0.0593 | 0.0246 | 0.0014 | 10.5446 |
| DDS | 1 | 0.7494 | 1.8943 | 2.7958 | 2.5925 | 48.8016 |
| DIS | 1 | 0.7027 | 1.7942 | 2.6746 | 2.5026 | 34.0449 |
| CS DDS | 1 | 0.0593 | **0.0573** | **0.0239** | **0.0012** | **9.3640** |
| CS DIS | 1 | **0.0331** | 0.0641 | 0.0244 | 0.0017 | 12.8663 |

However, at NFE=1 and NFE=2, the performance of DDS and DIS degrades considerably, with much higher Sinkhorn distances, especially for more complex datasets like rings and image. CS consistently outperforms both DDS and DIS in these few-step generation tasks, exhibiting significantly lower Sinkhorn distances. This is particularly evident in tasks like the double well distribution, where CS is as good as its 200-steps teacher with only one or two steps.

In Table 2, we measure the Sinkhorn distance between samples generated by the pre-trained diffusion-based samplers and their respective distillate consistency samplers. This experiment provides support for our theoretical analysis, demonstrating that the learned consistency sampler replicates the behavior of the teacher model.

Table 2: Sinkhorn distances between samples generated by the 200-step pre-trained diffusion-based samplers (DDS, DIS) and their corresponding 2-step distillated consistency samplers. The distilled consistency samplers closely replicate the performance of the teachers.

| Method | GMM | DW | Rings | Image |
|--------|-----|-----|-------|-------|
| CS vs DDS | 0.05725 | 0.00118 | 0.05747 | 0.02088 |
| CS vs DIS | 0.03310 | 0.00147 | 0.06585 | 0.02132 |

In summary, the results presented in both figures and Table 1 confirm that the consistency sampler enables faster generation than existing diffusion-based samplers. Notably, CS achieves one-step sampling, eliminating the need for iterative sampling.

## 6 CONCLUSION

In this work, we introduce consistency samplers, a new class of samplers designed for sampling from unnormalized distributions. Unlike most existing methods that require multiple iterative updates, consistency samplers can generate high-quality samples in one step. Consistency samplers amortize sampling from a pre-trained diffusion-based model by learning a direct mapping from any point along the sampling trajectory to the target distribution. This mapping enables one-step sampling from the target distribution, while retaining the flexibility to refine samples through multiple denoising and noise addition steps, offering to trade computational cost for accuracy.

A key advantage of our method is that it does not require access to pre-collected datasets. Rather than fully integrating the diffusion trajectories of a pre-trained diffusion-based sampler, our method learns the single-step mapping directly from intermediate noisy samples, reducing the training time.

Our experiments demonstrate that consistency samplers perform well in both one-step and few-step sampling tasks, achieving results comparable to diffusion-based samplers that require hundreds of steps, while maintaining good sample quality.

Obtaining samples under limited computational budgets remains a significant challenge. We see consistency samplers as a step toward more practical and efficient sampling, accelerating the application of sampling methods in large-scale and resource-constrained machine learning and scientific problems.

## 7 ETHICS STATEMENT

We adhere to the ICLR Code of Ethics and confirm that our experiments use only public datasets. While our results are primarily based on synthetic data, we recognize the potential for misuse and encourage responsible application of our methods on real-world data. We welcome any related discussions and feedback.

## 8 REPRODUCIBILITY STATEMENT

We provide detailed algorithmic and experimental description in Section 5, and we have an open sourced code with configuration files accompanying this research in this GitHub repository.

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

## A    DETAILS ABOUT DIFFUSION-BASED SAMPLERS

In this section, we provide detailed derivations of the training objectives for the denoising diffusion sampler (DDS) (Vargas et al., 2023) and the time-reversed diffusion sampler (DIS) (Berner et al., 2024). This presentation closely follows the original formulations of DDS and DIS, as well as the insightful unification presented by Richter & Berner (2024), which frames both approaches under the unified perspective of measures on path spaces and time-reversals of controlled stochastic processes.

### A.1    DENOISING DIFFUSION SAMPLER

The denoising diffusion sampler (DDS), introduced by Vargas et al. (2023), adopts the settings $\mu(\mathbf{x}_t, t) = -\beta_t \mathbf{x}_t$ and $\sigma(t) = \sigma\sqrt{2\beta_t}$, which correspond to the variance preserving (VP) SDE as described by Song et al. (2021b). In DDS, the control function $u_{\boldsymbol{\theta}} = \sigma(t)s_{\boldsymbol{\theta}}$ is used, where $s_{\boldsymbol{\theta}}$ is a neural network with parameters $\boldsymbol{\theta}$, designed to approximate the intractable score function in Eq. (2).

The objective of DDS is to solve the problem described by Eq. (3), where the divergence measure $D$ is the KL divergence. By applying the chain rule for the KL divergence, we can express the objective as:

$$D_{\mathrm{KL}}\left(\mathbb{P}_{\boldsymbol{\theta}} \| \mathbb{P}_{\mathbf{y}^v}\right) = D_{\mathrm{KL}}\left(p_{\mathrm{prior}} \| p_{\mathbf{y}_T}\right) + \mathbb{E}_{\mathbf{x}_0}\left[\mathbb{P}_{\boldsymbol{\theta}}(\cdot | \mathbf{x}_0) \| \mathbb{P}_{\mathbf{y}^v}(\cdot | \mathbf{x}_0)\right]$$

where $\mathbb{P}_{\boldsymbol{\theta}}$ denotes the path space measure of $\mathbf{x}^{u_{\boldsymbol{\theta}}}$.

Next, using Girsanov's theorem, the KL divergence over the path measures can be rewritten as:

$$D_{\mathrm{KL}}\left(\mathbb{P}_{\boldsymbol{\theta}} \| \mathbb{P}_{\mathbf{y}^v}\right) = D_{\mathrm{KL}}\left(p_{\mathrm{prior}} \| p_{\mathbf{y}_T}\right) + \sigma^2 \mathbb{E}_{\mathbb{P}_{\boldsymbol{\theta}}}\left[\int_0^T \beta_t \| s_{\boldsymbol{\theta}}(\mathbf{x}_t, t) - \nabla \log p_{\mathbf{y}_{T-t}}(\mathbf{x}_t)\|^2 \, \mathrm{d}t\right]. \quad (9)$$

However, the expectation in Eq. (9) still contains the intractable score function, making direct optimization difficult.

To address this issue, DDS introduces a reference inference process $\mathbf{y}^{\mathrm{ref}}$ that follows the same SDE, but initialized from a Gaussian distribution $p_{\mathbf{y}_0^{\mathrm{ref}}} = \mathcal{N}(0, \sigma^2 \boldsymbol{I})$ instead of the target distribution $p_{\mathrm{target}}$. This ensures that all marginals satisfy $p_{\mathbf{y}_t^{\mathrm{ref}}} = \mathcal{N}(0, \sigma^2 \boldsymbol{I})$, and in particular $\nabla \log p_{\mathbf{y}_t^{\mathrm{ref}}}(\mathbf{x}) = -\mathbf{x}/\sigma^2$.

This allows the KL divergence between the path measures to be rewritten as:

$$D_{\mathrm{KL}}\left(\mathbb{P}_{\boldsymbol{\theta}} \| \mathbb{P}_{\mathbf{y}^v}\right) = D_{\mathrm{KL}}\left(\mathbb{P}_{\boldsymbol{\theta}} \| \mathbb{P}_{\mathrm{ref}}\right) + \mathbb{E}_{\mathbf{x}_0}\left[\log \frac{p_{\mathbf{y}_0^{\mathrm{ref}}}(\mathbf{x}_0)}{p_{\mathbf{y}_0}(\mathbf{x}_0)}\right],$$

Where $\mathbb{P}_{\mathrm{ref}}$ denotes the path measure of $\mathbf{y}^{\mathrm{ref}}$.

The Radon-Nikodym derivative allows us to express the difference between the process $\mathbb{P}_{\boldsymbol{\theta}}$ and the reference process $\mathbb{P}_{\mathrm{ref}}$ as:

$$\log \frac{\mathrm{d}\mathbb{P}_{\boldsymbol{\theta}}}{\mathrm{d}\mathbb{P}_{\mathrm{ref}}} = \sigma^2 \int_0^T \beta_t \| s_{\boldsymbol{\theta}}(\mathbf{x}_t, t) + \mathbf{x}/\sigma^2\|^2 \, \mathrm{d}t + \sigma \int_0^T \sqrt{2\beta_t}\left(s_{\boldsymbol{\theta}}(\mathbf{x}_t, t) + \mathbf{x}/\sigma^2\right)^\top \mathrm{d}\mathbf{w}.$$

By combining the above expressions for the KL divergence and the Radon-Nikodym derivative, we arrive at the following DDS loss function:

$$\mathcal{L}_{\mathrm{DDS}} = \mathbb{E}_{\mathbb{P}_{\boldsymbol{\theta}}}\left[\sigma^2 \int_0^T \beta_t \| s_{\boldsymbol{\theta}}(\mathbf{x}_t, t) + \mathbf{x}/\sigma^2\|^2 \, \mathrm{d}t + \log \frac{\mathcal{N}(\mathbf{x}_0; 0, \sigma^2 \boldsymbol{I})}{\rho(\mathbf{x}_0)}\right] \quad (10)$$

This final objective enables DDS to avoid relying on the intractable score function by using the reference process, simplifying the optimization problem.

### A.2    TIME-REVERSED DIFFUSION SAMPLER

Using the representation of the Radon-Nikodym derivative, the time-reversed diffusion sampler (Berner et al., 2024) directly considers minimizing the divergence $D_{\mathrm{KL}}\left(\mathbb{P}_{\mathbf{x}^u} \| \mathbb{P}_{\mathbf{y}^v}\right)$. In practice,

the loss for DIS is formulated as:

$$\mathcal{L}_{\text{DIS}} = (\mathbb{P}_{\mathbf{x}^u} \| \mathbb{P}_{\mathbf{y}^v}) = \mathbb{E}\left[ \int_0^T \left( \frac{1}{2}\|u(\mathbf{x}_t^u, t)\|^2 - \operatorname{div}\mu(\mathbf{x}_t^u, t) \right) dt + \log \frac{p_{\text{prior}}(\mathbf{x}_0)}{\rho(\mathbf{x}_T^u)} \right]. \quad (11)$$

See the verification Theorem 2.4 in Berner et al. (2024) and Proposition 2.3 on the likelihood of path measures in Richter & Berner (2024).

## B  PROOF OF THEOREM 1

**Theorem 1.** *Let $\boldsymbol{f}_{\boldsymbol{\theta}}(\mathbf{x}_t, t)$ be a consistency sampler parameterized by $\boldsymbol{\theta}$, and let $\boldsymbol{f}(\mathbf{x}_t, t; u)$ denote the consistency function of the PF ODE defined by the control $u$. Assume that $\boldsymbol{f}_{\boldsymbol{\theta}}$ satisfies a Lipschitz condition with constant $L > 0$, such that for all $t \in [0, T]$ and for all $\mathbf{x}_t, \mathbf{y}_t$,*

$$\|\boldsymbol{f}_{\boldsymbol{\theta}}(\mathbf{x}_t, t) - \boldsymbol{f}_{\boldsymbol{\theta}}(\mathbf{y}_t, t)\|_2 \le L\|\mathbf{x}_t - \mathbf{y}_t\|_2.$$

*Additionally, assume that for each step $n \in \{1, 2, \ldots, N-1\}$, the ODE solver called at $t_n$ has a local error bounded by $O((t_{n+1} - t_n)^{p+1})$ for some $p \ge 1$.*

*If, additionally, $\mathcal{L}(\boldsymbol{\theta}, \boldsymbol{\theta}; u) = 0$, then:*

$$\sup_{n, \mathbf{x}_{t_n}} \|\boldsymbol{f}_{\boldsymbol{\theta}}(\mathbf{x}_{t_n}, t_n) - \boldsymbol{f}(\mathbf{x}_{t_n}, t_n; u)\|_2 = O((\Delta t)^p),$$

*where $\Delta t := \max_{n \in \{1,2,\ldots,N-1\}} |t_{n+1} - t_n|$.*

*Proof.* The proof is similar to the one presented by Song et al. (2023), with the key difference that we must account for the global integration error introduced by the ODE solver.

If the ODE solver, when called at $t_{n+1}$, has a local error uniformly bounded by $O((t_n - t_{n-1})^{p+1})$, then the cumulative error across all steps is approximately the sum of $n+1$ local errors and is bounded by $O((\Delta t)^p)$.

We are interested in $\boldsymbol{e}_n$, the error between the learned consistency sampler and the consistency function of the PF ODE defined by the control $u$ at $\mathbf{x}_{t_n} \sim p_{t_n}(\mathbf{x}_{t_n})$,

$$\boldsymbol{e}_n := \boldsymbol{f}_{\boldsymbol{\theta}}(\mathbf{x}_{t_n}, t_n) - \boldsymbol{f}(\mathbf{x}_{t_n}, t_n; u).$$

If $\mathcal{L}(\boldsymbol{\theta}, \boldsymbol{\theta}; u) = 0$, we deduce that

$$\lambda(t_n) d(f_{\boldsymbol{\theta}}(\hat{\mathbf{x}}_{t_{n+1}}^u, t_{n+1}), f_{\boldsymbol{\theta}}(\hat{\mathbf{x}}_{t_n}^u, t_n)) = 0.$$

Since $\lambda(t_n) > 0$, this implies:

$$\boldsymbol{f}_{\boldsymbol{\theta}}(\hat{\mathbf{x}}_{t_{n+1}}^u, t_{n+1}) = \boldsymbol{f}_{\boldsymbol{\theta}}(\hat{\mathbf{x}}_{t_n}^u, t_n). \quad (12)$$

We can derive a recurrence relation for $\boldsymbol{e}_n$:

$$\begin{aligned}
\boldsymbol{e}_n &\overset{(i)}{=} \boldsymbol{f}_{\boldsymbol{\theta}}(\mathbf{x}_{t_n}, t_n) - \boldsymbol{f}_{\boldsymbol{\theta}}(\hat{\mathbf{x}}_{t_n}^u, t_n) + \boldsymbol{f}_{\boldsymbol{\theta}}(\hat{\mathbf{x}}_{t_n}^u, t_n) - \boldsymbol{f}(\mathbf{x}_{t_{n+1}}, t_{n+1}; u) \\
&\overset{(ii)}{=} \boldsymbol{f}_{\boldsymbol{\theta}}(\mathbf{x}_{t_n}, t_n) - \boldsymbol{f}_{\boldsymbol{\theta}}(\hat{\mathbf{x}}_{t_n}^u, t_n) + \boldsymbol{f}_{\boldsymbol{\theta}}(\hat{\mathbf{x}}_{t_{n+1}}^u, t_{n+1}) - \boldsymbol{f}(\mathbf{x}_{t_{n+1}}, t_{n+1}; u) \\
&= \boldsymbol{f}_{\boldsymbol{\theta}}(\mathbf{x}_{t_n}, t_n) - \boldsymbol{f}_{\boldsymbol{\theta}}(\hat{\mathbf{x}}_{t_n}^u, t_n) + \boldsymbol{f}_{\boldsymbol{\theta}}(\hat{\mathbf{x}}_{t_{n+1}}^u, t_{n+1}) - \boldsymbol{f}_{\boldsymbol{\theta}}(\mathbf{x}_{t_{n+1}}, t_{n+1}) \\
&\qquad + \boldsymbol{f}_{\boldsymbol{\theta}}(\mathbf{x}_{t_{n+1}}, t_{n+1}) - \boldsymbol{f}(\mathbf{x}_{t_{n+1}}, t_{n+1}; u) \\
&= \boldsymbol{f}_{\boldsymbol{\theta}}(\mathbf{x}_{t_n}, t_n) - \boldsymbol{f}_{\boldsymbol{\theta}}(\hat{\mathbf{x}}_{t_n}^u, t_n) + \boldsymbol{f}_{\boldsymbol{\theta}}(\hat{\mathbf{x}}_{t_{n+1}}^u, t_{n+1}) - \boldsymbol{f}_{\boldsymbol{\theta}}(\mathbf{x}_{t_{n+1}}, t_{n+1}) + \boldsymbol{e}_{n+1} \\
&\cdots \\
&\overset{(iii)}{=} \boldsymbol{f}_{\boldsymbol{\theta}}(\mathbf{x}_{t_n}, t_n) - \boldsymbol{f}_{\boldsymbol{\theta}}(\hat{\mathbf{x}}_{t_n}^u, t_n) + \boldsymbol{f}_{\boldsymbol{\theta}}(\mathbf{x}_T, T) - \boldsymbol{f}_{\boldsymbol{\theta}}(\hat{\mathbf{x}}_T^u, T) + \boldsymbol{e}_T.
\end{aligned}$$

Here, step $(i)$ follows from the definition of the consistency function, step $(ii)$ is due to Eq. (12), and step $(iii)$ leverages the telescoping nature of the sum.

Furthermore, since $\boldsymbol{f_\theta}$ is parameterized such that $\boldsymbol{f_\theta}(\mathbf{x}_T, T) = \mathbf{x}_T$, we have

$$e_T = \boldsymbol{f_\theta}(\mathbf{x}_T, T) - \boldsymbol{f}(\mathbf{x}_T, T; u)$$
$$= \mathbf{x}_T - \mathbf{x}_T$$
$$= 0.$$

Finally, given that $\boldsymbol{f_\theta}$ is Lipschitz and considering the bound on the global error of the ODE solver:

$$\|\boldsymbol{e}_n\|_2 \leq \|\boldsymbol{e}_T\|_2 + L\|\mathbf{x}_{t_n} - \hat{\mathbf{x}}^u_{t_n}\|_2 + L\|\mathbf{x}_T - \hat{\mathbf{x}}^u_T\|_2 = O((\Delta t)^p).$$

$\square$

