# OpenReview forum: "Skip the Steps: Data-Free Consistency Distillation for Diffusion-based Samplers"
_ICLR.cc/2025/Conference — ICLR 2025 Conference Withdrawn Submission_

### Official Review · Reviewer_yJz9 · 2024-10-24

**Soundness:** 2
**Presentation:** 3
**Contribution:** 1
**Rating:** 3
**Confidence:** 4

**Summary:**

This paper presents a distillation framework based on consistency models, for general diffusion samplers (e.g., DDS, DIS).
Compared to consistency models for content generation, the most challenge in this setting is the absence of datasets (from the unnormalized target distributions).
To distill a pre-trained diffusion sampler, the authors first simulate training data from the prior distribution and then optimize the consistency distillation objective function, while some architectures are specially designed to reduce the computational cost.
The experiments support the claim of the faster sampling speed of consistency samplers, especially the one-step sampling quality.
This paper is clearly written and easy to follow, but I still have some questions and critiques as detailed below.

**Strengths:**

- This paper is clearly written and easy to follow.
- The problem considered in this paper is interesting and may inspire the following works. As far as I know, the authors are the first to consider distill the diffusion samplers which target at an unnormalized distribution.
- The proposed distillation approach seems straightforward but practical, and its performance is evaluated on several data sets.

**Weaknesses:**

- Minor contribution. I cannot see a significant difficulty in directly applying consistency distillation (Song,2023) to distilling pre-trained diffusion samplers. In my point of view, the most important ingredients of the distillation sampler, including the data-prediction matching idea, the distillation loss in Eq (9), and the sampling scheme, all come form the CM paper (Song, 2023). I admit there are several minor differences, e.g., the way to obtain the training data (N-step simulation from prior, instead of T-step), but these are natural choices with the absence of a dataset.
- The experiments are insufficient to demonstrate the superiority of distillation sampler. The target distributions in section 5.1 are all two-dimensional, and the $d$ for DW is not specified. Exploration on high-dimensional and complicated target distributions is necessary. Some examples include the $\phi^4$ model in (https://arxiv.org/abs/2402.10758) and the VAE model in (https://arxiv.org/pdf/2310.02679).
- The background spends much space in introducing the formulation and training objective of diffusion samplers, but this is not very related to the distillation problem, since a sampler is pre-given. I suggest to cut down the length of the background section.
- As the proposed method requires simulating the pre-given diffusion sampler for many steps, it naturally raises concerns about the computational cost in the training process. Adding more information on the computation time and memory would be better.

**Questions:**

- What does the $p_{target}$ mean in the equation of the Gaussian Mixture Model in section 5.1?

---

> ### Author Response · Authors · 2024-11-22
>
> - **Minor contribution.**
>   Thank you for your comment and for acknowledging the differences between our approach and consistency models. While it is true that our method shares some conceptual foundations with CMs, we believe that applying these ideas to enable single-step sampling for unnormalized distributions presents unique contributions.
>
>   One of the key novelties of our work is adapting the consistency distillation framework to operate without access to a dataset, $\mathcal{D}$, unlike CMs, which rely on sampling from $\mathcal{D}$ as described in algorithms 2 and 3 of the CM paper (Song, 2023). The method we propose leverages only a pre-trained diffusion-based sampler to perform N-step simulations from a prior, where $N$ is uniformly sampled. This approach allows consistency distillation to be applied in a broader range of scenarios where CMs would be constrained by the need for pre-existing data.
>
> - **Insufficient experiments.**
>   Thank you for your observation. In the revised manuscript, we have added results for a 100-dimensional double-well example to evaluate the performance of our method in a more challenging, high-dimensional setting. This result is included as a new column in Table 1 of the revised manuscript. The added results show that, in the 100-dimensional double-well setting, the consistency sampler (CS) achieves performance comparable to that of the 200-step teacher models (DDS and DIS) while requiring significantly fewer steps (1 or 2 NFEs).
>
>     | **Method** | **NFE** | **DW (d=100)** |
>     |------------|---------|----------------|
>     | DDS        | 200     | 10.9340        |
>     | DIS        | 200     | 10.9658        |
>     | DDS        | 2       | 30.5513        |
>     | DIS        | 2       | 20.7306        |
>     | CS DDS     | 2       | **9.4402**     |
>     | CS DIS     | 2       | 10.5446        |
>     | DDS        | 1       | 48.8016        |
>     | DIS        | 1       | 34.0449        |
>     | CS DDS     | 1       | **9.3640**     |
>     | CS DIS     | 1       | 12.8663        |
>
> - **Background.**
>   Thank you for your suggestion. In the revised manuscript, we have significantly reduced the length of the background section to focus on the most relevant details.
>
> - **Training computational costs.**
>   Thank you for pointing this out. We understand that simulating the given diffusion sampler during training could raise concerns about computational cost. However, the training cost of CS is less than that of the denoising diffusion sampler (DDS) and the time-reversed diffusion sampler (DIS). In DDS and DIS, the controlled process appears in the training objective (see equations (11) and (12)), necessitating the simulation of the entire trajectory and cannot resort to a Monte Carlo approximation, such as in the denoising score matching objective used in diffusion models [1]. Similarly, CS requires trajectory simulation during training; however, CS only integrates partial trajectories (as described in Algorithm 1) up to a random timestep. This saves approximately 50% of the training time, for a fixed number of training iterations.
>
>   We have added a discussion in Section 5 of the revised manuscript that provides a comparison of the training costs for our method relative to DDS and DIS.
>
>   1. Yang Song, Jascha Sohl-Dickstein, Diederik P. Kingma, Abhishek Kumar, Stefano Ermon, and Ben Poole. *Score-based generative modeling through stochastic differential equations*. In International Conference on Learning Representations, 2021.
>
> - **Meaning of $p_\text{target}$.**
>   This was a typo, and we removed it in the revised manuscript. Thank you for pointing it out.

---

> > ### Comment · Reviewer_yJz9 · 2024-11-26
> >
> > I thank the authors for their detailed clarification and the additional experiment. However, my following concerns remain:
> > - To the best of my knowledge, the double-well is not difficult to sample from, especially when the parameter $m$ (the most important parameter for the difficulty of the problem) is small. Notably, the authors do not explicitly state the choice of $m$. It is recommended to conduct some more challenging problems or discuss the limitations honestly, as the paper is a submission to a top-tier AI conference.
> > - The authors only discuss the time cost verbally, without reporting any quantitative results. Moreover, I don't think the time reduction of CS is 50%, because the samples are processed in batches and the random timesteps will cause inefficient parallelization.
> >
> > Based on these concerns, I will maintain my current score.

---

### Official Review · Reviewer_bXAe · 2024-10-31

**Soundness:** 2
**Presentation:** 3
**Contribution:** 2
**Rating:** 5
**Confidence:** 4

**Summary:**

This paper introduces "consistency samplers," a novel approach to generating high-quality samples from probability distributions in a single step, rather than requiring hundreds of iterative steps like traditional methods such as Markov Chain Monte Carlo (MCMC) and diffusion-based samplers. The key innovation is a consistency distillation algorithm that can learn to map directly from a simple prior distribution to a target distribution without needing access to training data or full sampling trajectories, making it much more computationally efficient.

The method works by distilling knowledge from pre-trained diffusion-based samplers, but does so efficiently by only using incomplete sampling trajectories and intermediate noisy representations. Rather than requiring full trajectory integration, the approach leverages probability flow ordinary differential equations (ODEs) to generate deterministic consecutive points for training. This allows the consistency sampler to learn the mapping between states while using roughly half the parameters of traditional diffusion-based samplers.

Through extensive experiments on multiple benchmark tasks, including Gaussian mixture models, double well potentials, ring distributions, and image data, the authors demonstrate that their method can achieve comparable or better sample quality compared to traditional approaches while being 100-200 times faster. They also provide theoretical guarantees showing that under certain conditions, the learned consistency sampler can approximate the true consistency function with arbitrary accuracy as the step size of the ODE solver decreases. This represents a significant advance in making sampling methods more practical and efficient for real-world applications.

**Strengths:**

The proposed method is novel, and is the first algorithm to propose to accelerate generation for unnormalized sampling problems. Further, this paper also gives a practical solution based on consistency models to achieve one-step generation. This provides flexibility to balance between speed and quality through optional refinement steps. The training doesn't require training data. It also avoids the need for full trajectory integration during training. It also provides theoretical guarantees for the method's convergence.

**Weaknesses:**

The algorithm is not entirely clear to me; for the setting of sampling from a given unnormalized density function, there is no appearance of the target density in Algorithm 1. I guess the whole algorithm might be first train with DDS method, and do consistency training based on the pretrained DDS model?

Another drawback is from the experiment side. Have you tried problems with 100+ / 1000+ dimensions like log Gaussian Cox process? Now most experiments are only in 2 dimensions.

**Questions:**

None.

---

> ### Author Response · Authors · 2024-11-22
>
> - **Algorithm 1.**
>   Thank you for your observation. You are correct that the target density does not explicitly appear in Algorithm 1. In our approach, we assume a pre-trained diffusion-based sampler is given, such as DDS, which already includes information about the normalized distribution. This setting is practical, given the availability of pre-trained models on platforms like GitHub and HuggingFace. Our method can significantly accelerate their convergence (e.g., achieving results in just one step) while maintaining comparable sampling performance.
>
> - **Numerical examples are low-dimensional.**
>   Thank you for your observation. In the revised manuscript, we have added results for a 100-dimensional double-well example to evaluate the performance of our method in a more challenging, high-dimensional setting. This result is included as a new column in Table 1 of the revised manuscript. The added results show that, in the 100-dimensional double-well setting, the consistency sampler (CS) achieves performance comparable to that of the 200-step teacher models (DDS and DIS) while requiring significantly fewer steps (1 or 2 NFEs).
>
>   | **Method** | **NFE** | **DW (d=100)** |
>   |------------|---------|----------------|
>   | DDS        | 200     | 10.9340        |
>   | DIS        | 200     | 10.9658        |
>   | DDS        | 2       | 30.5513        |
>   | DIS        | 2       | 20.7306        |
>   | CS DDS     | 2       | **9.4402**     |
>   | CS DIS     | 2       | 10.5446        |
>   | DDS        | 1       | 48.8016        |
>   | DIS        | 1       | 34.0449        |
>   | CS DDS     | 1       | **9.3640**     |
>   | CS DIS     | 1       | 12.8663        |

---

### Official Review · Reviewer_QxKa · 2024-10-31

**Soundness:** 2
**Presentation:** 2
**Contribution:** 1
**Rating:** 3
**Confidence:** 3

**Summary:**

Sampling of high-quality samples in time-constrained settings is a challenging problem. Consistency Models (CMs) allow for faster training and efficient sampling of diffusion models from complex probability distributions. The work builds on top of CMs and proposes a new class of samplers called Consistency Samplers (CS). CS distills representations from intermediate sampling steps along the diffusion process in order to learn a one-step mapping from any sampling step to the terminal step in the target distribution. This bypasses the need to generate and integrate complete sampling trajectories. CS enable one-step and multi-step sampling in order to trade-off computational cost for sample quality. Empirical evaluation on a range of probability distributions demonstrates faster samping when compared to prior diffusion-based samplers.

**Strengths:**

* The paper is well written and easy to understand.
* The work presents intuitive explanations with a theoretical result quantifying the gap betwen the learned and true consistency samplers.

**Weaknesses:**

* **Data-free sampling:** The paper claims that the proposed consistency sampler is data-free. However, the claim is not well supported as similar to Consistency Models (CMs) [1], the sampler is initialized at $x_{0} \sim p_{prior}$ which is a data sample. If I understand correctly, the key difference between CS and CM is that the sampler is executed for shorter run (i.e- terminated early) instead of running it upto full T sampling steps. While this requires less data in practice, the sampling process in itself cannot be cateogrized as data-free.
* **Similarity to CM:** A main concern from the paper is its contribution over the previously proposed CMs. It is unclear as to what is the additional contribution over CMs. CS terminates the sampling steps early and distills representations from intermediate samples. This can be inferred as a special case of CM sampling with shorter sampling steps. Even in the case of multi-step sampling, differences remain unclear as the training and sampling process remain exactly similar.
* **Comparison with CM:** While the authors compare CS to a range of prior diffusion sampling baselines, a clear comparison with CM is missing. Since the sampler directly borrows its training, distillation and short sampling steps from CMs, it would be a fair comparison to evaluate CS with CMs for shorter runs. Does CS provide high fidelity samples when compared to CMs? Can CS identify more modes early on when compared to CMs? What speedup and additional advantages does CS offer over prior CMs? A direct comparison with CMs will help answer these questions.
* **Benchmarks:** While the work presents empirical evaluation on a range of datasets, experiments are limited to small toy tasks with limited complexity and number of modes. CMs demonstrate benefits and performance improvements over prior methods when executed in higher dimensions with complex multi-modal data samples. Could the authors comment on how the sampler can be executed with high dimensional data such as images (CIFAR, ImageNet)? Experiments are not required, but an intuitive understanding of the sampler and its applicability to other tasks would benefit the paper as these modalities require samplers to be conditioned on data and run for longer sampling steps.

[1]. Song, Yang, et al. "Consistency models." arXiv preprint arXiv:2303.01469 (2023).

**Questions:**

Please refer to weaknesses section above.

---

> ### Author Response · Authors · 2024-11-22
>
> - **Data-free sampling.**
>   Thank you for your observation. To clarify, during training, consistency models (CMs) first sample $x \sim \mathcal{D}$, where $\mathcal{D}$ is a dataset of samples, and not from the prior distribution (as described in their algorithms 2 and 3). In contrast, our approach assumes that we are provided only with a pre-trained sampler and that no dataset $\mathcal{D}$ is available for training.
>
>   While it is true that our method involves partially simulating the PF ODE trajectory using the pre-trained sampler, this does not equate to using data from a dataset. The training process leverages the pre-trained model directly, without needing to pre-collect and store a large dataset, as sampling from the prior is easy. This differs from CMs, where training relies on access to a dataset. We believe that this nuanced characteristic supports our claim of being data-free.
>
> - **Similarity to CM and comparison with CM.**
>   Consistency sampler (CS) and consistency model (CM) are designed to address different tasks. CS tackles the classical sampling problem, where the goal is to sample from a target density $p_\text{target}:= \rho / Z$, with an intractable normalizing constant $Z$. In this setting, we assume access to an unnormalized density $\rho$ but no data sampled from $p_\text{target}$.
>
>   Applying CMs to this problem would first require converting the sampling task into a data generation task. This involves generating a dataset using a pre-trained sampler and then applying either consistency distillation or consistency training to learn the consistency model. In contrast, CS is specifically designed to avoid this intermediate step. By leveraging partial integration during training, CS bypasses the need to pre-collect and store a dataset, reducing both memory overhead and integration error by avoiding full trajectory integration.
>
> - **Benchmarks.**
>   Thank you for your thoughtful comment. To apply sampling methods like ours to high-dimensional data such as images (e.g., CIFAR or ImageNet), we would need access to an unnormalized probability distribution $\rho$ rather than training data. As mentioned previously, CS tackles the classical sampling problem, where the objective is to sample from a target density $\rho = p_\text{target}Z$. This setting differs from the data generation task addressed by diffusion models, which rely on access to training datasets.
>
>   The added results of a 100-dimensional double-well example in the revised manuscript demonstrate that our approach can handle higher-dimensional settings effectively. This example provides insight into the scalability of CS to more complex and higher-dimensional tasks.
>
>   | **Method** | **NFE** | **DW (d=100)** |
>   |------------|---------|----------------|
>   | DDS        | 200     | 10.9340        |
>   | DIS        | 200     | 10.9658        |
>   | DDS        | 2       | 30.5513        |
>   | DIS        | 2       | 20.7306        |
>   | CS DDS     | 2       | **9.4402**     |
>   | CS DIS     | 2       | 10.5446        |
>   | DDS        | 1       | 48.8016        |
>   | DIS        | 1       | 34.0449        |
>   | CS DDS     | 1       | **9.3640**     |
>   | CS DIS     | 1       | 12.8663        |

---

> > ### Comment · Reviewer_QxKa · 2024-11-25
> > **Response to Rebuttal**
> >
> > I thank the authors for their response. However, my concerns regarding the comparison with CM remain unaddressed.
> >
> > **Comparison with CM:** Note that CS and CM both leverage consistency training with the same distillation loss, i.e- by design CMs are capable of sampling from a target density. While this requires access to data samples, it is still a fair comparison since both methods rely on the same algorithmic desiderata. Can the authors compare CS to a naive CM which samples from the target distribution using shorter runs?

---

### Official Review · Reviewer_9nTx · 2024-11-03

**Soundness:** 2
**Presentation:** 3
**Contribution:** 2
**Rating:** 3
**Confidence:** 3

**Summary:**

This work considers pre-trained diffusion-based models. Rather than simulating from the controlled diffusion -- which is costly, especially for fine time discretisations -- the authors propose to amortise this problem by learning so called "consistency" mappings which map samples from any point along the trajectory to the target distribution.

**Strengths:**

1. **Clarity.** The writing is mostly clear and concise and thus relatively easy to read.

2. **Originality.** To my knowledge, this method is novel but I am not an expert on diffusion-based models.

3. **Correctness.** The method overall seems justified (but note the remarks about Theorem 1 below).

**Weaknesses:**

**Main comments:**

1. **Computational cost of training the consistency function.** If I understand Table 1 correctly, the claimed "100--200$\times$" speedup does not account for the cost of training the consistency function $f_\theta$. If this is correct, then I think the claims of speedups must be substantially softened. More generally, it would seem to me that the proposed methodology requires simulating a large number of trajectories according to the probability-flow ODE from Equation 8. However, once we have sampled a large number of trajectories according to (8), aren't we already done? I mean, then we already have samples from the target, no? [yes, Algorithm 1 only uses potentially partial trajectories but the computational complexity is the same].

2. **Assumptions in Theorem 1.** Do we not need additional assumptions on $f_\theta$ in Theorem 1? It seems to me that the stated assumptions would be satisfied for /any/ Lipschitz function $f_\theta$? For instance, wouldn't the assumptions hold for $f_\theta \equiv \mathrm{const}$ being some constant function? Or are there additional assumptions buried in the terminology "consistency sampler"/"consistency mapping"? If so, these should be stated here more clearly.

3. **Numerical examples are low-dimensional.** The method is applied only to fairly simple examples which all seem to be two-dimensional (except for the "double-well" example for which I did not find a mention of the dimension). I think there should be higher-dimensional examples.



**Other comments:**

* Is there a difference between "consistency sampler" and "consistency function" (e.g., in Theorem 1 or Section 4.5)? The former is never defined. Likewise, the term "consistency mapping of the PF ODE defined by the control $u$" in Theorem 1 is never defined.



**Typos:**

* L117: Grammar in "$T$-step MCMC transition"

* Eq. 1--3: Is "$\mathbb{w}$" missing a subscript?

* Eq. 8: missing punctuation

* Algorithm 2: Should $n$ be $N$ instead?

* Bibliography: In many places, proper nouns are not capitalised.

**Questions:**

1. How realistic is the Lipschitz assumption in Theorem 1? Can you give examples in which this is verifiable?

2. What was the dimension $d$ in the double-well example?

---

> ### Comment · Reviewer_9nTx · 2024-11-22
>
> I have now read the other reviews. In light of this, I am keeping my score.

---

> ### Author Response · Authors · 2024-11-22
>
> - **Computational cost of training the consistency function.**
>   Thank you for raising this point. While it is true that we can generate samples directly using the pre-trained diffusion-based sampler, our work explores whether it is feasible to distill this model into a faster sampler. The training cost of our consistency sampler (CS) is a one-time expense, which amortizes the sampling cost at inference time. This trade-off is advantageous for applications where the sampler will be repeatedly used to generate new samples, like molecular design [1] and graph optimization [2].
>
>   The training cost of CS is less than that of the denoising diffusion sampler (DDS) and the time-reversed diffusion sampler (DIS). In DDS and DIS, the controlled process appears in the training objective (see equations (11) and (12)), necessitating the simulation of the entire trajectory. Similarly, CS requires trajectory simulation during training; however, CS only integrates partial trajectories (as described in Algorithm 1) up to a random timestep, sampled uniformly. This saves approximately 50% of the training time for a fixed number of training iterations. We have added a discussion in Section 5 of the revised manuscript that provides a comparison of the training costs for our method relative to DDS and DIS.
>
>   It is important to note that the speed comparison in Table 1 refers to the generative process and does not account for training times. DDS typically requires between 64 and 500 steps for generation [3], whereas CS achieves high-quality results in just 1 or 2 steps, with minimal quality trade-offs. This comparison method aligns with the distillation literature, where test-time efficiency improvements are emphasized (see, for example, Table 2 in [4]).
>
>   1. Zhu, Y., Wu, J., Hu, C., Yan, J., Hsieh, C.-Y., Hou, T., and Wu, J. Sample-efficient multi-objective molecular optimization with GFlowNets. *Advances in Neural Information Processing Systems*, 2023.
>
>   2. Zhang, D., Dai, H., Malkin, N., Courville, A., Bengio, Y., and Pan, L. Let the Flows Tell: Solving Graph Combinatorial Optimization Problems with GFlowNets. *Proceedings of the Neural Information Processing Systems*, 2023.
>
>   3. Francisco Vargas, Will Sussman Grathwohl, and Arnaud Doucet. Denoising diffusion samplers. *International Conference on Learning Representations*, 2023.
>
>   4. Tim Salimans and Jonathan Ho. Progressive distillation for fast sampling of diffusion models. *International Conference on Learning Representations*, 2022.
>
> - **Assumptions in Theorem 1.**
>   No, we do not need additional assumptions. The assumption that the loss is optimized to zero is crucial, and we have explicitly mentioned this in the paper on lines 335-336. A constant function would not satisfy this requirement due to the boundary condition $f(x_T^u, T) = x_T^u$ imposed by the parameterization in Eq. (6). This boundary condition ensures that the consistency sampler correctly maps the terminal state to itself, thereby disqualifying trivial solutions such as constant functions.
>
> - **Consistency function, mapping, and sampler.**
>   By “consistency mapping” and “consistency function,” we were referring to the function that maps any intermediate state of a trajectory directly to its terminal state. The consistency sampler is a model representation of this function. To enhance clarity, we have standardized the terminology throughout the revised manuscript: we no longer use the term “consistency mapping.” The terms “consistency function" and “consistency sampler" are defined in Section 4.1.
>
> - **Numerical examples are low-dimensional.**
>   Thank you for your observation. In the revised manuscript, we have added results for a 100-dimensional double-well example to evaluate the performance of our method in a more challenging, high-dimensional setting. This result is included as a new column in Table 1 of the revised manuscript. The added results show that, in the 100-dimensional double-well setting, the consistency sampler (CS) achieves performance comparable to that of the 200-step teacher models (DDS and DIS) while requiring significantly fewer steps (1 or 2).
>
> | **Method** | **NFE** | **DW (d=100)** |
> |------------|---------|----------------|
> | DDS        | 200     | 10.9340        |
> | DIS        | 200     | 10.9658        |
> | DDS        | 2       | 30.5513        |
> | DIS        | 2       | 20.7306        |
> | CS DDS     | 2       | **9.4402**     |
> | CS DIS     | 2       | 10.5446        |
> | DDS        | 1       | 48.8016        |
> | DIS        | 1       | 34.0449        |
> | CS DDS     | 1       | **9.3640**     |
> | CS DIS     | 1       | 12.8663        |
>
> - **Typos.**
>   Thank you for pointing out typos. We fixed them in the revised version.

---

> ### Author Response · Authors · 2024-11-22
>
> - **Lipschitz assumption.**
>   Thank you for raising this question. The Lipschitz assumption in Theorem 1 is a standard regularity condition used to ensure that the consistency function behaves predictably with respect to small changes in its input. This assumption is also made for consistency models.
>
> - **Double-well dimension.**
>   Thank you for pointing this out. In the double-well example, the dimension was d=2. We have now clarified this in the revised manuscript. Additionally, we have included results for a higher-dimensional double-well example with d=100.

---

### Note · Authors · 2024-12-04

**Comment:**

We sincerely appreciate the reviewers for their feedback. Given the short rebuttal period, we are unable to address all the concerns raised in the reviews comprehensively. As a result, we have decided to take more time to carefully revise our work and withdraw the current submission.

**Withdrawal Confirmation:**

I have read and agree with the venue's withdrawal policy on behalf of myself and my co-authors.